# Neutropenic sepsis and septic shock in ICU patients: A single-center experience over the last decade

Florian Guillotin[1], Laetitia Aubert[2], Soraya Benguerfi[1,3], Reyes Munoz Calahorro[1], Alice Vennier[1], David Boutoille[4], Thomas Gastinne[5], Jean Reignier[1,3], Jean-Baptiste Lascarrou[1,3], Matilde Karakachoff[2], Emmanuel Canet[1]*

1 Service de Médecine Intensive Réanimation, CHU de Nantes, Nantes Université, Nantes, France, 2 Nantes Université, CHU Nantes, Pôle Hospitalo-Universitaire 11: Santé Publique, Clinique des données, INSERM, Nantes, France, 3 Intensive Care Unit, Nantes University Hospital, Movement-Interactions-Performance Research Unit (MIP, UR 4334), Nantes, France, 4 Service de Maladies Infectieuses, CHU de Nantes, Nantes Université, Nantes, France, 5 Service d'Hématologie, CHU de Nantes, Nantes Université, Nantes, France

* emmanuel.canet@chu-nantes.fr

## Abstract

### Purpose

Sepsis and septic shock in patients with neutropenia are associated with high mortality. We investigated the features and outcome predictors of neutropenic sepsis in the last decade.

### Methods

Consecutive patients who were admitted to the intensive care unit (ICU) of a French university-affiliated hospital in 2012–2022, met criteria for sepsis or septic shock, and had neutropenia were included retrospectively. Patient features were collected and compared for 2012–2017 and 2018–2022. Factors associated with hospital mortality were sought by univariate and multivariate analyses.

### Results

Of the 185 patients, 85 were admitted in 2012–2017 and 100 in 2018–2022. The more recent group was older and had a heavier comorbidity burden but had a hospital mortality rate of 40.0% compared to 49.4% in the early group (p = 0.24). The most common source infections were pulmonary (24.8%) and hepatobiliary or gastrointestinal (23.8%). Gram-negative bacilli predominated. Predictors of in-hospital mortality were older age (odds ratio [OR], 1.04; 1.01–1.07; P = 0.005) and worse SOFA score (OR, 1.22; 1.05–1.42; P = 0.009). Aminoglycoside therapy predicted lower in-hospital mortality regardless of infection site and renal function (OR, 0.30; 0.14–0.63;

**Data availability statement:** The raw data underlying our findings contain sensitive personal health information that may potentially allow for patient re-identification. In accordance with the French data protection law, as enforced by the Commission Nationale de l'Informatique et des Libertés (CNIL), we are legally prohibited from making these data publicly available. Researchers may request access to the data for ethically approved scientific purposes by contacting the corresponding author: emmanuel.canet@chu-nantes.fr.

**Funding:** The author(s) received no specific funding for this work.

**Competing interests:** EC has received lecturer and speaker fees, as well as reimbursements of travel and accommodation expenses related to attending scientific meetings, from Gilead, Shionogi, and Sanofi-Genzyme. JBL has received lecturer and conference-speaker fees from BD and Zoll. None of the other authors have any financial or non-financial interests to disclose. This does not alter our adherence to PLOS One policies on sharing data and materials.

$P = 0.002$). Early source-control interventions were not significantly associated with hospital mortality.

## Conclusion

Intensivists are admitting patients with neutropenic sepsis who are older and have more comorbidities than was the case in earlier years. Despite this change, there is a non-statistically significant trend of declining mortality. Our findings support the initiation at ICU admission of combination antibiotic therapy including an aminoglycoside.

## Introduction

The number of patients living with cancer is increasing due to life-extending advances in cancer treatment [1] combined with aging of the population [2]. Malignancies and their treatments compromise the immune system, thereby creating a risk of life-threatening complications. Unplanned intensive-care-unit (ICU) admission has been reported in 5.2% of patients with solid tumors [3] and 13.9% of those with hematological malignancies [4]. Sepsis is one of the main reason for ICU admission in patients with cancer, notably those with neutropenia [5]. Progression to septic shock is associated with mortality rates of 40% to 60% [6,7].

The unique immunologic profile of neutropenic patients often blunts typical inflammatory responses, complicating early diagnosis of sepsis. Although current guidelines for managing septic shock apply to patients with cancer [8,9], there's still limited insight into the particularities of neutropenic sepsis. Indeed, the existing observational data are limited by inconsistent use of the most recent definitions of sepsis and septic shock, and inclusion of heterogeneous patient groups (e.g., clinically diagnosed sepsis or non-neutropenic cancer patients). The unique characteristics of neutropenic sepsis includes different sources of infection (e.g., mucositis, catheters, neutropenic enterocolitis), a broader spectrum of pathogens, an evolving epidemiology of antimicrobial resistance, and a greater susceptibility to acute kidney injury (AKI) when compared to patients without neutropenia [10,11]. Conceivably, a better knowledge of neutropenic sepsis in the recent era may contribute to guiding clinical practice and improving patients' outcomes.

The primary objective of this retrospective observational cohort study in a single ICU was to define the features and outcomes of neutropenic patients with sepsis or septic shock over the last decade. The secondary objective was to identify predictors of hospital mortality. We hypothesized that survival will have improved over time despite increases in age and comorbidity burden.

## Methods

This study was approved by the ethics committee of the French Intensive Care Society (CE SRLF 23–070) on August 14, 2023. In accordance with French law on retrospective observational studies of anonymized healthcare data, informed consent was not required. This report complies with STROBE guidelines (Supplementary Appendix, S1 Table).

## Study design, setting, and population

We retrospectively identified adults admitted to the ICU of the Nantes University Hospital for neutropenic sepsis or septic shock between January 1, 2012, and December 31, 2022. Only the first admission was considered in patients admitted more than once during the study period. The Nantes University Hospital is a 2432-bed public hospital with a high level of activity and expertise for managing patients having various conditions associated with immunodeficiency (solid organ transplantation, hematological malignancies, and solid cancers). The ICU is a closed, 30-bed, medical unit that admits 1600–1800 patients per year, of whom about one-third have immunodeficiency. At our institution, senior intensivists are available on site 24 h a day/7 days a week and immunocompromised patients admitted to the ICU are jointly managed by the ICU team and specialists in hematology, solid cancer, solid organ transplantation, infectious diseases, and general surgery.

Patients were identified by searching the electronic hospital database for patients assigned both the code for neutropenia (D70) and any of the codes for sepsis (A400-403, A409, A415, A419) and/or septic shock (R572) in the International Classification of Diseases-tenth revision. Each medical file was identified by the use of the local biomedical data warehouse [12] was reviewed by FG to confirm the diagnosis of neutropenic sepsis or septic shock. The other inclusion criteria were age ≥ 18 years; ICU admission with suspected infection as the main reason; hemodynamic instability requiring continuous vasopressor infusion to maintain the mean arterial pressure ≥65 mm Hg; and neutropenia defined as a neutrophil count <500/mm$^3$ or a leukocyte count <1000/mm$^3$. Patients with shock not due to sepsis and patients without neutropenia were not included. Quinolone prophylaxis is not used in our hospital and 3 different types of central lines are used for cancer patients: peripherally inserted central catheter (PICC) lines, totally implanted catheter (port), and tunneled externalized catheters (TEC).

## Data collection and outcomes

For each patient, the data reported in Tables 1 and 2 were extracted from the electronic hospital and ICU databases (CERNER Millenium®, North Kansas City, MI; and IntelliSpace Critical Care & Anesthesia, Philips®, Amsterdam, The Netherlands; respectively). The primary investigator (FG) manually extracted the data after a detailed review of each medical file. The site of infection was reported as mentioned by the intensivist in charge in the patients' formal discharge summary. Central lines-related infections were defined according to the criteria recommended by the French Intensive Care Society [13].

We divided the patients into early and recent groups defined by admission during the six years between January 1, 2012, and December 31, 2017; and the five years between January 1, 2018, and December 31, 2022; respectively.

We collected the Sequential Organ Failure Assessment (SOFA) score and the Simplified Acute Physiology Score II (SAPS II) on the day after ICU admission. Sepsis and septic shock were defined according to Sepsis-3 criteria [14]. Bloodstream infection (BSI) was defined according to the CDC/NHSN criteria [15]. Primary BSI was defined as bacteremia with an unidentified site of infection and secondary BSI as bacteremia associated with a documented site of infection. In the case of BSI due to skin commensal bacteria, two blood cultures with identical antibiotic susceptibility test results were required to confirm the diagnosis when no other source of infection was identified. Early source control was defined as performance of central lines removal within 24 hours after ICU admission and/or any image-guided procedure or surgical intervention targeting the source of sepsis within 48 hours after ICU admission (S1 File).

All aspects of sepsis and septic-shock management were at the discretion of the intensivists in charge, who followed best-practice guidelines, notably regarding the initial choice of antibiotic therapy and decisions to remove intravascular devices [8,15–17]. Rapid diagnostic microbiologic tests were not used in our ICU during the study period.

Vital status was collected at ICU and hospital discharge and 90 days after ICU admission. The day-90 data were extracted from the electronic hospital database.

**Table 1. Baseline characteristics of the 185 study participants.**

| Variables | ICU admission in 2012–2017 (n = 85) | ICU admission in 2018–2022 (n = 100) | *P* value |
|---|---|---|---|
| **Age, years, median [IQR]** | 59.00 [46.00–68.00] | 64.00 [52.00–69.25] | 0.057 |
| **Females, n (%)** | 30 (35.3) | 34 (34.0) | 0.88 |
| **Charlson Comorbidity Index, median [IQR]** | 3.50 [2.00–5.00] | 4.00 [3.00–5.00] | 0.055 |
| **Clinical Frailty Scale score, median [IQR]** | 4.00 [3.00–5.00] | 3.00 [2.00–4.00] | 0.003 |
| **ECOG Performance Status, n (%)** | | | 0.19 |
| 0 or 1 | 61 (71.8) | 67 (67.0) | |
| 2 | 18 (21.2) | 23 (23.0) | |
| 3 or 4 | 6 (7.1) | 10 (10.0) | |
| **Primary malignancy, n (%)** | | | |
| Solid tumor | 15 (17.6) | 17 (17.0) | >0.99 |
| Acute myeloid leukemia | 20 (23.5) | 29 (29.0) | 0.50 |
| Acute lymphoid leukemia | 12 (14.1) | 4 (4.0) | 0.018 |
| Multiple myeloma | 4 (4.7) | 4 (4.0) | >0.99 |
| Lymphoma | 19 (22.3) | 34 (34.0) | 0.10 |
| **Autologous HSCT, n (%)** | 10 (11.8) | 9 (9.0) | 0.63 |
| **Allogeneic HSCT, n (%)** | 15 (17.6) | 20 (20.0) | 0.71 |
| **CAR-T cell therapy, n (%)** | 0 (0.0) | 3 (3.0) | 0.25 |
| **Targeted therapy, n (%)** | 1 (1.2) | 8 (8.0) | 0.04 |
| **Severity scores** | | | |
| SOFA score, median [IQR] | 10.00 [9.00–11.00] | 9.00 [7.00–11.00] | 0.04 |
| SAPS II, median [IQR] | 56.00 [46.00–72.00] | 57.00 [47.75–73.25] | 0.92 |
| **Clinical data at ICU admission** | | | |
| GCS, median [IQR] | 15.00 [15.00–15.00] | 15.00 [15.00–15.00] | 0.89 |
| On room air, n (%) | 18 (21.2) | 43 (43.0) | 0.002 |
| Invasive mechanical ventilation | 7 (8.2) | 4 (4.0) | 0.35 |
| **Laboratory data, median [IQR]** | | | |
| Lactate at ICU admission, mmol/L, missing data n = 16 | 2.40 [1.67–3.90] | 2.30 [1.50–4.30] | 0.92 |
| Highest lactate during ICU stay, mmol/L, missing data n = 2 | 3.20 [1.80–5.80] | 3.30 [1.83–6.00] | 0.96 |
| Serum creatinine at ICU admission, µmol/L | 109.00 [83.00–159.00] | 106.00 [80.50–151.25] | 0.53 |

CAR, chimeric antigen receptor; ECOG, eastern cooperative oncology group; GCS, glasgow coma scale; HSCT, hematopoietic stem-cell transplantation; IQR, interquartile range; SAPS II, simplified acute physiology score version II; SOFA, sepsis-related organ failure assessment.

## Objectives

The primary objective of the study was to compare the clinical features and outcomes (Tables 1 and 2) of patients admitted to the ICU for neutropenic sepsis or septic shock between the two time periods (2012–2017 and 2018–2022).

The secondary objective was to identify factors associated with hospital mortality.

## Statistical analysis

Continuous variables are described as median [interquartile range] and compared using Wilcoxon's signed rank test. Categorical variables are reported as counts (percentage) and compared using the exact Fisher's test. Missing data were counted. To identify variables associated with hospital mortality, we performed logistic regression to estimate odds ratios (ORs) with their 95% confidence intervals (95%CIs). We built a multivariable model using variables that were either associated with hospital mortality in previous studies (age, SOFA score, site of

**Table 2. Sites of infection and microbiological documentation.**

| Variables | ICU admission in 2012–2017 (n = 85) | ICU admission in 2018–2022 (n = 100) | *P* value |
|---|---|---|---|
| **Site of primary infection,ᵃ n (%)** | | | |
| Pulmonary | 26 (30.6) | 20 (20.0) | 0.12 |
| Abdominal | 15 (17.6) | 29 (29.0) | 0.08 |
| Central line | 7 (8.2) | 8 (8.0) | >0.99 |
| Skin and soft tissues | 4 (4.7) | 7 (7.0) | 0.55 |
| Other | 7 (8.2) | 8 (8.0) | >0.99 |
| Primary BSI | 19 (22.4) | 13 (13.0) | 0.12 |
| Unidentified | 14 (16.5) | 20 (20.0) | 0.57 |
| **Secondary BSI, n (%)** | 17 (20.0) | 39 (39.0) | 0.006 |
| **Multiple sites of infection, n (%)** | 7 (8.2) | 5 (5.0) | 0.39 |
| **Microbiological documentation,ᵇ n (%)** | | | |
| **Bacteria** | 65 (76.5) | 70 (70.0) | |
| Gram-negative bacilli | *41 (48.2)* | *54 (54.0)* | *0.46* |
| Gram-positive cocci | *22 (25.9)* | *13 (13.0)* | *0.04* |
| *Clostridium difficile* | *2 (2.4)* | *3 (3.0)* | *1.00* |
| **Fungi** | 6 (7.0) | 2 (2.0) | |
| *Aspergillus* spp. | *3 (3.5)* | *1 (1.0)* | 0.34 |
| *Candida* spp. | *3 (3.5)* | *1 (1.0)* | 0.34 |
| **Viruses** | 4 (4.7) | 4 (4.0) | |
| Influenza virus | *3 (3.5)* | *0 (0.0)* | 0.10 |
| Enterovirus | *1 (1.2)* | *0 (0.0)* | 0.46 |
| SARS-Cov-2 | *0 (0.0)* | *4 (4.0)* | 0.13 |
| **No microorganisms recovered, n (%)** | 23 (27.1) | 31 (31.0) | 0.63 |

BSI, bloodstream infection; SARS-Cov-2, severe acute respiratory syndrome-coronavirus-2.

ᵃIn both groups, some patients had more than one site of primary infection.

ᵇSome patients in both groups had more than one type of microorganism identified.

infection, and serum creatinine) [18–20] or plausibly associated with hospital mortality (aminoglycoside therapy, early source control, and early vs. recent time period). Multicollinearity among the variables in the multivariate logistic model was checked using the Variance Inflation Factor (VIF). All VIF values were between 1.02 and 1.13, suggesting low collinearity among the explanatory variables. All tests were two-sided, and *P* values lower than 5% were considered to indicate significant associations. All analyses were performed using the R program version 4.3.2 (https://www.r-project.org).

## Results

### Study population

Of the 185 included patients, 54 (29.2%) had sepsis without shock and 131 (70.8%) had septic shock. Table 1 reports their main features. Time from neutropenia onset to ICU admission was 2 [0–7] days and neutropenia duration was 7 [3–16] days. Patients admitted during the most recent period were older and had more comorbidities, although the differences were not statistically significant. General health was good in both groups with a performance status of 0–2 in 91% of cases overall. The Clinical Frailty Scale score was lower in the recent than in the early group. The lower SOFA score and higher proportion of patients on room air in the recent period suggest earlier ICU admission during the course of sepsis, although lactatemia was similar in the two groups.

 

## Characteristics and management of sepsis

The most common sources of sepsis were pulmonary (24.8%) and hepatobiliary or gastrointestinal (23.8%) (Table 2). Of the 134 patients with central lines, 67 (50%) had the device removed within 24 hours following ICU admission. However, central lines-related infections accounted for less than 10% of cases. The source of sepsis remained undetermined in nearly 20% of the patients. The proportion of patients who required chemotherapy at ICU admission was significantly lower in the recent period.

Before ICU admission, 12 (6.5%) patients had a history of infection with a resistant pathogen (extended-spectrum β-lactamase [ESBL]-producing Gram-negative bacillus [GNB], n = 11; and methicillin-resistant *Staphylococcus aureus*, n = 1). A pathogen was identified in 70% of the cases, and GNBs were the most commonly identified pathogens. Resistant pathogens were found in 8 (4.3%) patients (ESBL-producing GNB, n = 6; AmpC-producing Enterobacteral, n = 1; and multidrug-resistant *Pseudomonas aeruginosa*, n = 1). The prevalence of Gram-positive cocci was significantly lower during the recent period than during the early period.

Anti-pseudomonal broad-spectrum antibiotic therapy was given to most patients, usually in combination with an aminoglycoside (Table 3). The number of patients treated with granulocyte colony-stimulating factor (G-CSF) was higher in the recent period than in the early period. Leukocyte transfusions were used in only two patients, both with necrotizing skin and soft-tissue infections. Source-control interventions were performed in 40% of the patients. The duration of vasopressor infusion was not significantly different between the two periods, whereas invasive mechanical ventilation and renal replacement therapy were used significantly less often in the recent period. Serum creatinine at ICU admission did not differ significantly between the two groups.

## Factors associated with hospital mortality

Variables associated with higher hospital mortality by univariate analysis were older age, greater number and severity of organ dysfunctions (defined by a higher SOFA score), pulmonary source of infection, higher serum creatinine at ICU admission, and higher lactatemia (Table 4). Aminoglycoside therapy and failure to identify the source of sepsis were associated with lower hospital mortality.

By multivariate analysis, older age and greater number and severity of organ dysfunctions at ICU admission (defined by a higher SOFA score) were associated with higher hospital mortality, whereas combination aminoglycoside therapy was associated with lower hospital mortality (Fig 1, the model AUROC was 0.766).

# Discussion

## Key findings

In this retrospective observational cohort study of neutropenic patients with sepsis or septic shock and requiring ICU admission, we found several differences between the early six-year period (2012–2017) and the recent five-year period (2018–2022). In the recent period, the patients were older and had more comorbidities. Importantly, hospital mortality was nearly 20% lower in the recent period, though this improvement isn't statistically significant. Older age and worse organ dysfunction at ICU admission (defined by a higher SOFA score) were independently associated with higher hospital mortality. On the opposite, an aminoglycoside as part of the antibiotic regimen prescribed at ICU admission was independently associated with lower hospital mortality.

## Comparison to previous studies

Few studies have focused on neutropenic patients with cancer and sepsis or septic shock. Neutropenic sepsis has been reported chiefly in patients with hematological malignancies, who contributed 83% of our population [21–23]. Major advances in cancer treatment and supportive care over the past two decades have contributed to broaden the criteria for

**Table 3. ICU management and outcomes.**

| Variables | ICU admission in 2012–2017 (n = 85) | ICU admission in 2018–2022 (n = 100) | P value |
|---|---|---|---|
| **Antimicrobial treatment on day 1, n (%)** | | | |
| Anti-pseudomonal broad-spectrum antibiotic therapy[a] | *83 (97.6)* | *94 (94.0)* | 0.29 |
| Carbapenem | *31 (36.5)* | *34 (34.0)* | 0.76 |
| Aminoglycoside | *64 (75.3)* | *69 (69.0)* | 0.41 |
| Antibiotic active against aerobic gram-positive cocci | *38 (44.7)* | *29 (29.0)* | 0.02 |
| Fluoroquinolone | *5 (5.9)* | *1 (1.0)* | 0.10 |
| Antifungal agent[b] | *44 (51.8)* | *44 (44)* | 0.31 |
| Appropriate empiric antibiotic therapy, n (%)[c] | 63 (87.5) | 62 (89.9) | 0.79 |
| G-CSF, n (%) | 33 (38.8) | 54 (54.0) | 0.05 |
| Chemotherapy, n (%) | 7 (8.2) | 1 (1.0) | 0.03 |
| Red-blood-cell packs, median [IQR] | 2.00 [0.00–5.00] | 2.00 [0.00–4.00] | 0.39 |
| Platelet units, median [IQR] | 2.00 [1.00–6.00] | 3.00 [1.00–6.00] | 0.70 |
| **Source-control procedures, n (%)** | | | |
| Intravascular catheter removal within 24 hours | 34 (60.7) | 33 (41.2) | 0.036 |
| Surgery within 48 hours | 6 (7.1) | 10 (10.0) | 0.60 |
| **Life support** | | | |
| Vasopressor infusion, days, median [IQR] | 2.00 [1.00–4.25] | 2.00 [1.00–3.00] | 0.28 |
| Invasive mechanical ventilation, n (%) | 50 (58.8) | 40 (40.0) | 0.012 |
| Renal replacement therapy, n (%) | 20 (23.5) | 11 (11.0) | 0.030 |
| **Outcomes** | | | |
| ICU LOS, days, median [IQR] | 5.00 [3.00–11.00] | 4.00 [2.00–7.00] | 0.090 |
| Hospital LOS, days, median [IQR] | 15.00 [5.00–39.00] | 20.50 [6.00–33.50] | 0.50 |
| ICU mortality, n (%) | 38 (44.7) | 33 (33.0) | 0.13 |
| Hospital mortality, n (%) | 42 (49.4) | 40 (40.0) | 0.24 |
| Day-90 mortality, n (%) | 44 (51.8) | 45 (45.0) | 0.38 |

G-CSF, granulocyte colony-stimulating factor; ICU, intensive care unit; LOS, length of stay.

[a]any of the following: cefepime, piperacillin-tazobactam, and carbapenem.

[b]this variable combines antifungal prophylaxis, empirical antifungal treatment and targeted treatment.

[c]defined as in vitro pathogen susceptibility to the antibiotic regimen given on the first ICU day, in those patients with microbiological documentation of the infection.

ICU admission of patients with cancer [24,25]. The older age and heavier comorbidity burden of admitted patients seen in our study during the recent period vs. the early period corroborates previous findings [26–28].

In a retrospective study of 428 critically-ill patients with severe sepsis or septic shock, pulmonary and abdominal infections predominated, in keeping with our findings [26]. The initial infection escaped identification in nearly 20% of our patients, as reported previously [29]. An increase over time in GNB infections among neutropenic patients with cancer and sepsis has been reported [29]. GNB infections predominated in our cohort but did not increase significantly between the early and recent periods. Multidrug-resistant pathogens contributed fewer than 5% of the organisms recovered in our patients compared to nearly 25% in a 2010–2017 study from Spain confined to patients with hematological malignancies [30], reflecting differences in epidemiology between the two countries.

The impact of neutropenia on mortality in critically-ill cancer patients remains unclear [21]. Several studies have reported better survival over time in critically-ill neutropenic patients with cancer and sepsis or septic shock [6]. Similarly, in our cohort, hospital mortality was nearly 20% lower in 2018–2022 than in 2012–2017, despite the older age and higher

**Table 4. Univariate logistic regression analysis for factors associated with hospital mortality.**

| Factors | OR (95%CI) | *P* value |
|---|---|---|
| **Age, per additional year** | 1.03 (1.01–1.06) | 0.004 |
| **Charlson Comorbidity Index[a]** | 1.04 (0.88–1.23) | 0.66 |
| **Clinical Frailty Score[a]** | 1.09 (0.90–1.32) | 0.36 |
| **SOFA score on day 1[a]** | 1.25 (1.11–1.42) | <0.001 |
| **Early source-control procedure, yes/no** | 0.63 (0.34–1.13) | 0.13 |
| **Site of infection** | | |
| Pulmonary | 2.56 (1.26–5.35) | 0.011 |
| Identified and not pulmonary | ref | |
| Not identified | 0.40 (0.16–0.94) | 0.043 |
| **Aminoglycoside plus at least one other antimicrobial, yes/no** | 0.42 (0.22–0.81) | 0.010 |
| **Serum creatinine at ICU admission** | 1.00 (1.00–1.01) | 0.035 |
| **Highest lactate during ICU stay[b]** | 1.24 (1.13–1.39) | <0.001 |
| **Time period** | | |
| 2012–2017 | ref | |
| 2018–2022 | 0.68 (0.38–1.22) | 0.20 |

CI, confidence interval; OR, odds ratio; SOFA, sepsis-related organ failure assessment.

[a]per additional point.

[b]per additional mmol/L.

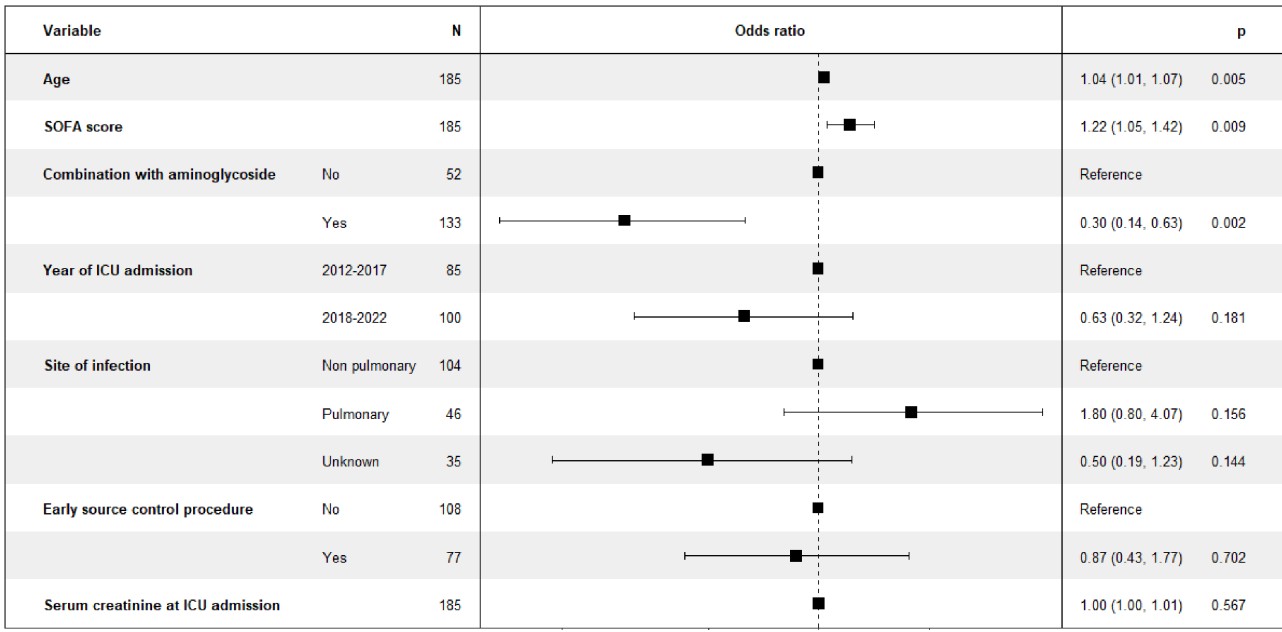

**Fig 1. Multivariable logistic regression analysis to identify factors associated with hospital mortality.**

comorbidity index in the recent group. Older age and higher SOFA score at ICU admission were independently associated with higher hospital mortality, as reported previously [22,23,26].

The best empirical antibiotic therapy for patients with neutropenic sepsis remains debated. Aminoglycoside therapy has been suggested for patients with hemodynamic instability, but the underlying level of evidence is low [9,16]. Meta-analyses have produced conflicting results and raised concern about the renal toxicity of aminoglycosides [31–34]. However, two retrospective studies suggest better survival with combination antibiotic therapy including an aminoglycoside [26,30]. In our cohort, aminoglycoside therapy combined with at least one other antimicrobial agent was independently associated with significantly lower hospital mortality.

Patients with neutropenia often have a central line. Early central line removal is recommended by French guidelines in the event of suspected infection or confirmed sepsis from an unidentified source [35]. In our study, half the patients with a central line had the device removed within 24 hours after ICU admission and the proportion was significantly smaller in the recent period. Early source control involving central line removal and/or surgery was not associated with better hospital survival in our cohort or in previous studies [36,37].

Granulocyte colony-stimulating factor (G-CSF) was used more often during the recent than the early period. G-CSF therapy shortens the duration of neutropenia. A meta-analysis suggested an association of G-CSF therapy with lower mortality in patients with neutropenic sepsis [21]. G-CSF therapy was not associated with hospital mortality in our cohort. A study of ICU patients with neutropenia also found no survival benefit with G-CSF therapy but did not focus on sepsis [38].

## Study implications

Our study confirms that reported improvements in the survival of critically ill patients with cancer extend to the sub-group with neutropenia and sepsis or septic shock. Whether these improvements are related to better patient selection to ICU admission and/or to therapeutic advances cannot be determined from our study. The lower SOFA score and higher proportion of patients on room air in the recent period suggest that ICU admission may be occurring earlier. Conceivably, earlier admission may contribute to better survival. Older age and comorbidities should not lead to refusal of ICU admission provided the patient is in good general health. Our results support the inclusion of an aminoglycoside in the initial antibiotic regimen, regardless of the infection site and renal function. The lower hospital mortality with combination aminoglycoside therapy was found despite a low prevalence of multidrug-resistant pathogens in our study, supporting a synergistic effect with other antibiotics. Finally, the low frequency of central line infection and absence of an association between early source-control procedures and hospital mortality suggest that leaving central lines in place may be warranted in many patients. Central line removal may be necessary only when central line infection is strongly suggested by physical findings or documented microbiologically.

## Strengths and limitations

A major strength of our study is the collection of data over an 11-year period with a comparison of two time periods. The retrospective design is a limitation of our study. No conclusion can be drawn about whether the association of combination aminoglycoside therapy with lower hospital mortality reflects a causal link. Large randomized controlled trials of this point are needed. Second, we were unable to determine the duration until appropriate antimicrobial therapy, which may influence mortality. Third, recruitment was at a single ICU. Our findings may not apply to all ICUs. Differences in case-mix and ICU-admission policies may affect hospital mortality of patients with cancer and neutropenic sepsis. Similarly, the variations in bacterial ecology across hospitals and ICUs should be borne in mind when considering our results about combination aminoglycoside therapy.

## Conclusion

In our ICU, the features of patients admitted for neutropenic sepsis or septic shock have changed in recent years. The patients are older and have a heavier burden of comorbidities but are in similar general health and have less severe organ dysfunction at ICU admission. While not statistically significant, hospital survival has shown improvement in recent years.

Aminoglycoside therapy in addition to at least one other antimicrobial agent for empirical therapy at ICU admission was associated with lower hospital mortality. Randomized trials are needed to further assess the effect of aminoglycoside therapy.

## Supporting information

**S1 Table. STROBE Statement.**
(DOC)

**S1 File. Definition of early source control.**
(DOC)

## Acknowledgments

**Consent to participate**

In accordance with French law on retrospective studies of anonymized healthcare data, informed consent was not required.

## Author contributions

**Conceptualization:** Florian Guillotin, Soraya Benguerfi, Reyes Munoz Calahorro, David Boutoille, Thomas Gastinne, Jean Reignier, Jean-Baptiste Lascarrou, Emmanuel Canet.

**Data curation:** Florian Guillotin, Soraya Benguerfi, Reyes Munoz Calahorro, Alice Vennier, Jean-Baptiste Lascarrou.

**Formal analysis:** Florian Guillotin, Laetitia Aubert, Matilde Karakachoff, Emmanuel Canet.

**Investigation:** Florian Guillotin, Jean Reignier.

**Methodology:** Florian Guillotin, Laetitia Aubert, Matilde Karakachoff, Emmanuel Canet.

**Software:** Laetitia Aubert, Matilde Karakachoff.

**Supervision:** Emmanuel Canet.

**Validation:** Alice Vennier, Emmanuel Canet.

**Visualization:** Florian Guillotin.

**Writing – original draft:** Florian Guillotin.

**Writing – review & editing:** Soraya Benguerfi, Reyes Munoz Calahorro, Alice Vennier, David Boutoille, Thomas Gastinne, Jean Reignier, Jean-Baptiste Lascarrou, Emmanuel Canet.

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
