## [Decision Letter · Decision Letter 0]

4 Jun 2025

PONE-D-25-18837Neutropenic Sepsis and Septic Shock in ICU patients: a single-center experience over the Last Decade.PLOS ONE

Dear Dr. Canet,

Thank you for submitting your manuscript to PLOS ONE. After careful consideration, we feel that it has merit but does not fully meet PLOS ONE’s publication criteria as it currently stands. Therefore, we invite you to submit a revised version of the manuscript that addresses the points raised during the review process.

We look forward to receiving your revised manuscript.

Kind regards,

Monia Marchetti

Academic Editor

PLOS ONE

Journal Requirements:

2.  Thank you for stating the following in the Competing Interests section: [EC has received lecturer and speaker fees, as well as reimbursements of travel and accommodation expenses related to attending scientific meetings, from Gilead, Shionogi, and Sanofi-Genzyme.

JBL has received lecturer and conference-speaker fees from BD and Zoll.

None of the other authors have any financial or non-financial interests to disclose.].

4. Please include captions for your Supporting Information files at the end of your manuscript, and update any in-text citations to match accordingly. Please see our Supporting Information guidelines for more information: http://journals.plos.org/plosone/s/supporting-information .

Reviewers' comments:

Reviewer's Responses to Questions

**Comments to the Author**

1. Is the manuscript technically sound, and do the data support the conclusions?

Reviewer #1: Yes

Reviewer #2: Partly

2. Has the statistical analysis been performed appropriately and rigorously? 

Reviewer #1: Yes

Reviewer #2: I Don't Know

3. Have the authors made all data underlying the findings in their manuscript fully available?

Reviewer #1: No

Reviewer #2: No

4. Is the manuscript presented in an intelligible fashion and written in standard English?

Reviewer #1: Yes

Reviewer #2: Yes

5. Review Comments to the Author

Reviewer #1: The authors investigated a topic of interest and the results of this research appears of relevant importance to inform the clinicians. However, I would authors to address the following issues:

- specify what is intended with "source control" an what you mean with "early"

- report the time to appropriate antimicrobial therapy and whether fast microbiology was used; this point would provide a further potential explanation for mortality improvement despite patients were apparently more severe in recent years;

- perform a multivariable regression analysis that include mechanical ventilation

Reviewer #2: Thank you for the opportunity to review this manuscript.

The authors have completed a single-centre retrospective cohort study of 185 patients with neutropenic sepsis admitted to intensive care. The describe clinical characteristics and changes in outcome between 2012-2017 and 2018-2022.

Recent studies on this topic are often registry-based or rely on population and administrative datasets, so I think the patient-level detail gathered in this study is a worthwhile addition to the literature.

My main concern with the manuscript is that the authors report quite a wide range of findings from a relatively small sample size (e.g. profile of infections, changes over time, impact of aminoglycosides); some of these findings reported were secondary outcomes that the study was not explicitly designed for and I feel they are treated a bit superficially. Specifically, the question of aminoglycoside use in neutropenic sepsis is an important one, but this is a complex question and was not the main aim of the study. There is a substantial body of literature on this topic (which the authors acknowledge in the discussion section) including some well-designed retrospective studies and meta-analyses. In order to meaningfully extend the existing research, I think a more dedicated analysis would be needed. There would need to be a stronger attempt to control for confounding factors such as causative organism, receipt of appropriate antibiotic therapy, timing of antibiotic administration etc, and I think this would need review from a statistical reviewer. The authors are careful not to state a causative relationship, but I still don’t think the quality of evidence produced justifies the emphasis on aminoglycoside use that is given in the manuscript. In any case, the main aim of the study was not to investigate aminoglycoside use and I think overall the manuscript would be improved by clearer focus on the primary objectives and the primary outcome - i.e. describing clinical characteristics and outcomes, and their temporal trends over the last 10 years.

I also think the framing of sepsis-3 criteria needs reconsidering. This is given a lot of emphasis in the manuscript. However, the authors report Sepsis-3 criteria were applied to included patients only after patient identification using a registry based on ICD codes (line 91 onward). Therefore it isn’t really accurate to say that patients were identified with Sepsis-3 alone. ICD codes are known to lack sensitivity and some patients excluded from the registry may not have been assessed against sepsis-3 criteria at all. Also, because Sepsis-3 criteria are poorly validated in neutropenic patients, it doesn’t automatically follow that using Sepsis-3 over other patient identification strategies is necessarily a better approach, and this seems to be taken as a given in the manuscript. I think it’s a reasonable approach, just not necessarily a key strength of the study per se. Therefore, if the use of Sepsis-3 is to be a key focus in the manuscript, I think it needs more nuanced exploration. Some useful references on the topic include the below:

Nathan, N., J.-P. Sculier, L. Ameye, M. Paesmans, G. Bogdan-Dragos, and A.-P. Meert, Sepsis and Septic Shock Definitions in Patients With Cancer Admitted in ICU. Journal of Intensive Care Medicine, 2021. 36(3): p. 255-261.

Valentine, Jake C., Karin A. Thursky, and Leon J. Worth. "Sepsis in cancer: a question of definition." Australian and New Zealand journal of public health 44.3 (2020): 245.

Cheng, Q., Y. Tang, Q. Yang, E. Wang, J. Liu, and X. Li, The prognostic factors for patients with hematological malignancies admitted to the intensive care unit. Springerplus, 2016. 5(1): p. 1-12.

Duke, Graeme J., et al. "Performance of hospital administrative data for detection of sepsis in Australia: The sepsis coding and documentation (SECOND) study." Health Information Management Journal 53.2 (2024): 61-67.

Henig, Oryan, et al. "The performance of sepsis-3 criteria to predict mortality among patients with hematologic malignancy and post-transplant who have suspected infection." Open Forum Infectious Diseases. 2021: 8(11)

Otherwise I have a few specific comments

Abstract

In both the abstract and the discussion section the authors comment on lower mortality in the second period but this difference was not statistically significant - this should be more clearly reported

Introduction:

The introduction is a bit short. I’m not entirely clear about what the authors are trying to express and how the existing literature has informed the design of the study. I.e. what are the key gaps in the existing literature that led to development of this study? Is it that studies of cancer patients use inconsistent sepsis definitions or that dedicated studies of neutropenic sepsis are lacking?

Regarding sepsis definitions, in line 54 the authors state “Few studies of patients with cancer used the Sepsis-3 criteria, and none focused on patients with neutropenia [9–11].”

Reference 9 did perform an analysis stratified by neutropenic status and reference 24 uses Sepsis-3 criteria (together with clinical criteria) in neutropenic patients. Perhaps it would suffice to say that existing observational data are limited by inconsistent use of definitions and inclusion of heterogenous patient groups (e.g. clinically diagnosed sepsis or non-neutropenic cancer patients).

Methods

Please supply some more detail about data extraction. Lines 103-4 “table 1 and 2 were extracted from the electronic hospital and ICU databases." Who performed the data extraction? Was there a detailed chart review by a clinician or were data extracted automatically? What definitions were used e.g. to determine infection site? What definition was used to classify CVC-related infections defined (ECDC, CDC/NHSN, clinician-determined, ICD codes, other?)? How were BSI defined and did common skin commensals require additional criteria for inclusion?

Statistical analysis: “missing data were counted” Are these reported anywhere in the manuscript?

Line 130 “Survival curves plotted using the Kaplan-Meier method for the early and recent periods were compared by applying the log-rank test” Is this reported anywhere in the results? It needs to be more clearly labelled if so. If it was performed, was this only univariable analysis?

For the logistic regression model, what model validation was used? How was multicollinearity assessed and managed? What was the model AUROC? (This could just be reported with the model in the results)

For the multivariable model, given that mechanical ventilation and renal replacement therapy differed between the two time periods should this be included in the model? Mechanical ventilation is consistently associated with increased mortality and seems important. Should receipt of appropriate antibiotic therapy be included in the model for the same reason?

Some background information on your centre would be valuable. How large is the hospital, is it a dedicated cancer centre? Is quinolone prophylaxis used? What type of central lines are used in cancer patients? Were there any changes to routine care during the study period (especially considering the COVID pandemic impacting period 2 and not period 1).

Results

Suggest in general avoid fractions like “three 5ths” (Line 320, line 391) and just give a percentage throughout.

Table 3 – the authors report receipt of antifungal as a risk factor but it needs to be clear if this was prophylaxis or treatment (i.e. it might include empiric antifungal for suspected fungal sepsis versus routine prophylaxis which would depend on underlying disease). If that can’t be established I don’t think it should be included in the model.

The authors discuss “number of organ failures” but actually seem to only include SOFA score in their model. This isn’t strictly speaking the same thing and should be either removed or analysed separately.

Discussion

Line 237 -238 “Few studies have focused on neutropenic patients with cancer and sepsis or septic shock defined by Sepsis-3 criteria [8].” Please refer to the comments above, again important that these were not the only criteria used in this study as only patients with ICD coded sepsis were considered for inclusion.

Line 249. “Multidrug-resistant pathogens contributed fewer than 5% of the organisms recovered in our patients compared to nearly 25% in a 2010–2017 study from Spain confined to patients with hematological malignancies [24,30].” Suggest removing this as this reflects known differences in epidemiology in these two countries, and seems like an arbitrary comparison. (Also reference 24 is not relevant).

Strengths section: Please see above about the use of sepsis-3 criteria as a strength. Line 298 “Third, we identified a factor (combination aminoglycoside therapy) independently associated with lower hospital mortality.” It’s not really usual to describe a finding of the study as a strength of the study. Please see above re: level of emphasis this finding is given.

Limitations: Given that the second period includes the COVID period and the first period does not, does this require any comments as a potential confounding factor? (E.g. was there any change in patient mix or processes during that period, is there any other evidence of impact on outcomes from cancer patients in other studies?)

6. PLOS authors have the option to publish the peer review history of their article (what does this mean? ). If published, this will include your full peer review and any attached files.

**Do you want your identity to be public for this peer review?** For information about this choice, including consent withdrawal, please see our Privacy Policy .

Reviewer #1: No

Reviewer #2: No

---

## [Author Response · Author response to Decision Letter 1]

18 Jul 2025

Reviewer #1

The authors investigated a topic of interest and the results of this research appears of relevant importance to inform the clinicians. However, I would authors to address the following issues:

Authors’ reply: We would like to thank the reviewer for making constructive comments that have helped us to clarify and improve our manuscript.

- specify what is intended with "source control" an what you mean with "early"

Authors’ reply: We agree with the reviewer that this important point needs to be clarified. Appropriate source control is a key principle in the management of sepsis and septic shock, in combination with appropriate antimicrobial therapy [1]. It also applies to neutropenic patients [2,3]. Source control refers to any procedure aimed at eradicating the source of sepsis. It includes removal of a potentially infected device (like central lines), or definitive control of a source of ongoing microbial : surgical intervention for the removal of intra-abdominal abscesses, treatment of gastrointestinal perforation, ischaemic bowel or volvulus, cholangitis, cholecystitis, pyelonephritis associated with obstruction or abscess, removal of necrotizing soft tissue infection, other deep space infection (e.g., empyema or septic arthritis), and implanted device infections.

It is recommended that source control should be achieved as soon as possible following initial resuscitation [1], yet there are limited data to conclusively issue a recommendation regarding the timeframe in which source control should be obtained. In our study, “Early” was pragmatically defined as central lines removal within 24 hours after ICU admission and/or any image-guided procedure or surgical intervention targeting the source of sepsis within 48 hours after ICU admission.

As requested by the reviewer, we added this information in the revised version of the manuscript (methods section and supplementary appendix).

1) Evans L et al. Intensive Care Med (2021) 47:1181–1247

2) Schnell D et al., Ann Intensive Care. 2016 Dec;6(1):90.

3) Kochanek M et al., Ann Hematol. 2019 May;98(5):1051-1069.

- report the time to appropriate antimicrobial therapy and whether fast microbiology was used; this point would provide a further potential explanation for mortality improvement despite patients were apparently more severe in recent years;

Authors’ reply: We agree that earlier initiation of effective antimicrobial therapy is associated with improved survival, as shown in a large retrospective cohort study (1). However, robust data from randomized controlled trials are still missing for neutropenic patients with sepsis. Moreover, determining with accuracy the onset of sepsis/septic shock is extremely challenging, as most of the patients are transferred in the ICU from the wards or the ED, where continuous monitoring is not feasible. Finally, the duration before appropriate antimicrobial therapy cannot be determined in patients without microbiological identification of a pathogen, a common situation in neutropenic patients. Unfortunately, we do not have the data of duration until appropriate antimicrobial therapy. We acknowledge this point in the limitation section of the revised version of the manuscript, as follows: “Moreover, we were unable to determine the duration until appropriate antimicrobial therapy, which may influence mortality”.

Fast microbiology is not used in current clinical practice in our department. This practice is restricted to clinical research studies. We added this information in the methods section as follows: “Rapid diagnostic microbiologic tests were not used in our ICU during the study period”.

- perform a multivariable regression analysis that include mechanical ventilation

Authors’ reply: We agree that invasive mechanical ventilation (IMV) is strongly associated with survival in neutropenic patients with septic shock. In our study, the use of IMV during the ICU stay is unsurprisingly associated with an increased risk of hospital mortality.

OR 95%CI P value

Invasive Mechanical Ventilation 6.98 3.51;14.36 <0.001

However, this variable is already captured in the respiratory component of the SOFA score (for patients with IMV implemented in the first 24 hours following ICU admission), which is included in our multivariable analysis. Finally, adding a variable with all patients treated with IMV is not possible in our multiple logistic regression analysis because IMV is a time-dependent variable.

Reviewer #2:

Thank you for the opportunity to review this manuscript.

The authors have completed a single-centre retrospective cohort study of 185 patients with neutropenic sepsis admitted to intensive care. The describe clinical characteristics and changes in outcome between 2012-2017 and 2018-2022.

Recent studies on this topic are often registry-based or rely on population and administrative datasets, so I think the patient-level detail gathered in this study is a worthwhile addition to the literature.

Authors’ reply: We would like to thank the reviewer for his thorough review of our manuscript, and for making constructive comments that have helped us to clarify and improve some important points.

My main concern with the manuscript is that the authors report quite a wide range of findings from a relatively small sample size (e.g. profile of infections, changes over time, impact of aminoglycosides); some of these findings reported were secondary outcomes that the study was not explicitly designed for and I feel they are treated a bit superficially. Specifically, the question of aminoglycoside use in neutropenic sepsis is an important one, but this is a complex question and was not the main aim of the study. There is a substantial body of literature on this topic (which the authors acknowledge in the discussion section) including some well-designed retrospective studies and meta-analyses. In order to meaningfully extend the existing research, I think a more dedicated analysis would be needed. There would need to be a stronger attempt to control for confounding factors such as causative organism, receipt of appropriate antibiotic therapy, timing of antibiotic administration etc, and I think this would need review from a statistical reviewer. The authors are careful not to state a causative relationship, but I still don’t think the quality of evidence produced justifies the emphasis on aminoglycoside use that is given in the manuscript. In any case, the main aim of the study was not to investigate aminoglycoside use and I think overall the manuscript would be improved by clearer focus on the primary objectives and the primary outcome - i.e. describing clinical characteristics and outcomes, and their temporal trends over the last 10 years.

Authors’ reply: We understand the reviewer’s comment and we agree. Regarding the specific concern about the aminoglycosides use, we comment on this finding from our secondary objective (i.e. identifying predictors of hospital mortality) for several reasons. First, it is the only modifiable factor associated with hospital mortality we identified. Second, the use of aminoglycosides in this setting remains a subject of debate owing to the low level of supporting evidence. Third, our findings complement those of other teams and support the need for a randomized trial on this topic.

We transparently report on our local epidemiology of pathogens (Table 2 and results section) and the appropriateness of initial empiric antibiotic therapy (Table 3). Analysing the timing of antibiotics requires the accurate determination of the onset of sepsis/septic shock, which is extremely challenging because most of the patients are transferred in the ICU from the wards or the ED, where continuous monitoring is not feasible. Moreover, the duration before appropriate antimicrobial therapy cannot be determined in patients without microbiological identification of a pathogen, a common situation in neutropenic patients. Unfortunately, we do not have the data of duration until appropriate antimicrobial therapy. We acknowledge this point in the limitation section of the revised version of the manuscript, as follows: “Second, we were unable to determine the duration until appropriate antimicrobial therapy, which may influence mortality”.

We acknowledge that no causality relationship can be determined from our retrospective single-centre experience. We also agree that our study, given its sample size, was not powered to definitively address all the findings of our secondary outcomes with the same rigor as studies specifically designed for those purposes. An attempt to account for confounders (which we believe is not feasible and would be misleading) would result in same inherent limitations of such retrospective analysis.

To address this point, we toned down and rephrased the relevant sections to present these findings more cautiously, clearly framing them as exploratory, descriptive observations rather than attempting to draw inferences for clinical practice. We ensured that our conclusions remained within the limitations of our study design and sample size.

I also think the framing of sepsis-3 criteria needs reconsidering. This is given a lot of emphasis in the manuscript. However, the authors report Sepsis-3 criteria were applied to included patients only after patient identification using a registry based on ICD codes (line 91 onward). Therefore it isn’t really accurate to say that patients were identified with Sepsis-3 alone. ICD codes are known to lack sensitivity and some patients excluded from the registry may not have been assessed against sepsis-3 criteria at all. Also, because Sepsis-3 criteria are poorly validated in neutropenic patients, it doesn’t automatically follow that using Sepsis-3 over other patient identification strategies is necessarily a better approach, and this seems to be taken as a given in the manuscript. I think it’s a reasonable approach, just not necessarily a key strength of the study per se. Therefore, if the use of Sepsis-3 is to be a key focus in the manuscript, I think it needs more nuanced exploration. Some useful references on the topic include the below:

Nathan, N., J.-P. Sculier, L. Ameye, M. Paesmans, G. Bogdan-Dragos, and A.-P. Meert, Sepsis and Septic Shock Definitions in Patients With Cancer Admitted in ICU. Journal of Intensive Care Medicine, 2021. 36(3): p. 255-261.

Valentine, Jake C., Karin A. Thursky, and Leon J. Worth. "Sepsis in cancer: a question of definition." Australian and New Zealand journal of public health 44.3 (2020): 245.

Cheng, Q., Y. Tang, Q. Yang, E. Wang, J. Liu, and X. Li, The prognostic factors for patients with hematological malignancies admitted to the intensive care unit. Springerplus, 2016. 5(1): p. 1-12.

Duke, Graeme J., et al. "Performance of hospital administrative data for detection of sepsis in Australia: The sepsis coding and documentation (SECOND) study." Health Information Management Journal 53.2 (2024): 61-67.

Henig, Oryan, et al. "The performance of sepsis-3 criteria to predict mortality among patients with hematologic malignancy and post-transplant who have suspected infection." Open Forum Infectious Diseases. 2021: 8(11)

Authors’ reply: We thank the reviewer for giving us the opportunity to clarify this point. The identification of patients did not exclusively rely on ICD-10 coding system, but also by cross-checking with a local biomedical data warehouse created and developed to ensure accuracy by extracting the data directly from the raw laboratory data (e.g. lactate) and the ICU clinical monitoring system (eg blood pressure, use and dose of norepinephrine,…) [1]. An ultimate checking of the medical files was done manually by the primary investigator (FG).

Nonetheless, we agree that this methodological aspect receives undue emphasis in the manuscript and cannot be considered as a strength by itself. We rephrased the introduction and the discussion sections of the manuscript to remove and/or soften the reference to Sepsis-3, which is no longer displayed as a strength in the revised version of the manuscript.

[1] Karakachoff M, Goronflot T, Coudol S, Toublant D, Bazoge A, Constant Dit Beaufils P, et al. Implementing a Biomedical Data Warehouse From Blueprint to Bedside in a Regional French University Hospital Setting: Unveiling Processes, Overcoming Challenges, and Extracting Clinical Insight. JMIR Med Inform. 2024;12: e50194. doi:10.2196/50194

Otherwise I have a few specific comments

Abstract

In both the abstract and the discussion section the authors comment on lower mortality in the second period but this difference was not statistically significant - this should be more clearly reported

Authors’ reply: Done as suggested. We now clearly state that the observed difference did not reach statistical significance.

Introduction:

The introduction is a bit short. I’m not entirely clear about what the authors are trying to express and how the existing literature has informed the design of the study. I.e. what are the key gaps in the existing literature that led to development of this study? Is it that studies of cancer patients use inconsistent sepsis definitions or that dedicated studies of neutropenic sepsis are lacking?

Regarding sepsis definitions, in line 54 the authors state “Few studies of patients with cancer used the Sepsis-3 criteria, and none focused on patients with neutropenia [9–11].”

Reference 9 did perform an analysis stratified by neutropenic status and reference 24 uses Sepsis-3 criteria (together with clinical criteria) in neutropenic patients. Perhaps it would suffice to say that existing observational data are limited by inconsistent use of definitions and inclusion of heterogenous patient groups (e.g. clinically diagnosed sepsis or non-neutropenic cancer patients).

Authors’ reply: As requested by the reviewer, we have revised the manuscript's introduction (second and third paragraphs) to better frame the study's rationale and objectives.

Methods

Please supply some more detail about data extraction. Lines 103-4 “table 1 and 2 were extracted from the electronic hospital and ICU databases." Who performed the data extraction? Was there a detailed chart review by a clinician or were data extracted automatically? What definitions were used e.g. to determine infection site? What definition was used to classify CVC-related infections defined (ECDC, CDC/NHSN, clinician-determined, ICD codes, other?)? How were BSI defined and did common skin commensals require additional criteria for inclusion?

Authors’ reply: As requested, we added more information about data extraction and definitions. The primary investigator manually extracted the data after a detailed review of each medical file. The site of infection was reported as mentioned by the intensivist in charge in the patients’ formal discharge summary. CVC-related infections were defined according to the criteria recommended by the French Intensive Care Society (1). Bloodstream infections were defined according to the CDC/NHSN criteria (2,3). In the case of bacteraemia due to skin commensal bacteria, two blood cultures with identical antibiotic susceptibility test results were required to confirm the diagnosis when no other source of infection was identified.

(1) Timsit JF, et al. Ann Intensive Care. 2020 Sep 7;10(1):118.

(2) Timsit JF, et al. Intensive Care Med (2020) 46:266–284

(3) https://www.cdc.gov/nhsn/pdfs/pscmanual/4psc_clabscurrent.pdf

Statistical analysis: “missing data were counted” Are these reported anywhere in the manuscript?

Authors’ reply: We had some missing data for Lactate at ICU admission (n=16) and highest lactate during ICU stay (n=2). As requested, we added this information in Table 1.

Line 130 “Survival curves plotted using the Kaplan-Meier method for the early and recent periods were compared by applying the log-rank test” Is this reported anywhere in the results? It needs to be more clearly labelled if so. If it was performed, was this only univariable analysis?

Authors’ reply: We apologize for this mistake. This analysis was not conducted. We removed the sentence in the revised version of the manuscript.

For the logistic regression model, what model validation was used?

---

## [Decision Letter · Decision Letter 1]

29 Sep 2025

Neutropenic Sepsis and Septic Shock in ICU patients: a single-center experience over the Last Decade.

PONE-D-25-18837R1

Dear Dr. Canet,

We’re pleased to inform you that your manuscript has been judged scientifically suitable for publication and will be formally accepted for publication once it meets all outstanding technical requirements.

Kind regards,

Monia Marchetti

Academic Editor

PLOS ONE

Additional Editor Comments (optional):

Reviewers' comments:

Reviewer's Responses to Questions

**Comments to the Author**

1. If the authors have adequately addressed your comments raised in a previous round of review and you feel that this manuscript is now acceptable for publication, you may indicate that here to bypass the “Comments to the Author” section, enter your conflict of interest statement in the “Confidential to Editor” section, and submit your "Accept" recommendation.

Reviewer #1: All comments have been addressed

2. Is the manuscript technically sound, and do the data support the conclusions?

Reviewer #1: Yes

3. Has the statistical analysis been performed appropriately and rigorously? 

Reviewer #1: Yes

4. Have the authors made all data underlying the findings in their manuscript fully available?

Reviewer #1: No

5. Is the manuscript presented in an intelligible fashion and written in standard English?

Reviewer #1: Yes

6. Review Comments to the Author

Reviewer #1: The authors have addressed my previous comments and the revised version of the manuscript appears significantly improve. I have no further comments.

7. PLOS authors have the option to publish the peer review history of their article (what does this mean? ). If published, this will include your full peer review and any attached files.

**Do you want your identity to be public for this peer review?** For information about this choice, including consent withdrawal, please see our Privacy Policy .

Reviewer #1: No

---

## [Editor Report · Acceptance letter]

PONE-D-25-18837R1

PLOS ONE

Dear Dr. Canet,

I'm pleased to inform you that your manuscript has been deemed suitable for publication in PLOS ONE. Congratulations! Your manuscript is now being handed over to our production team.

Kind regards,

on behalf of

Dr. Monia Marchetti

Academic Editor

PLOS ONE